# Defective Patient NK Function Is Reversed by AJ2 Probiotic Bacteria or Addition of Allogeneic Healthy Monocytes

**DOI:** 10.3390/cells11040697

**Published:** 2022-02-16

**Authors:** Meng-Wei Ko, Kawaljit Kaur, Tahmineh Safaei, Wuyang Chen, Christine Sutanto, Paul Wong, Anahid Jewett

**Affiliations:** 1The Jane and Jerry Weintraub Center for Reconstructive Biotechnology, Division of Oral Biology and Medicine, School of Dentistry, University of California Los Angeles, Los Angeles, CA 90095, USA; mengwei@g.ucla.edu (M.-W.K.); drkawalmann@g.ucla.edu (K.K.); tahimenh_saf@yahoo.com (T.S.); franklinechen0201@gmail.com (W.C.); sutanto.christine@gmail.com (C.S.); pwong5469@g.ucla.edu (P.W.); 2The Jonsson Comprehensive Cancer Center, UCLA School of Dentistry and Medicine, Los Angeles, CA 90095, USA

**Keywords:** NK cells, IFN-γ, monocytes, differentiation, AJ2, CD16 receptor

## Abstract

In this paper, we present the role of autologous and allogeneic monocytes from healthy individuals and those of the cancer patients, with a number of distinct cancers, in activating the function of natural killer (NK) cells, in particular, in induction of IFN-γ secretion by the NK cells and the functional capability of secreted IFN-γ in driving differentiation of the tumor cells. In addition, we compared the roles of CD16 signaling as well as sonicated probiotic bacteria AJ2 (sAJ2)-mediated induction and function of IFN-γ-mediated differentiation in tumor cells. We found that monocytes from cancer patients had lower capability to induce functional IFN-γ secretion by the autologous CD16 mAb-treated NK cells in comparison to those from healthy individuals. In addition, when patient monocytes were cultured with NK cells from healthy individuals, they had lower capability to induce functional IFN-γ secretion by the NK cells when compared to those from autologous monocyte/NK cultures from healthy individuals. Activation by sAJ2 or addition of monocytes from healthy individuals to patient NK cells increased the secretion of functional IFN-γ by the NK cells and elevated its functional capability to differentiate tumors. Monocytes from cancer patients were found to express lower CD16 receptors, providing a potential mechanism for their lack of ability to trigger secretion of functional IFN-γ. In addition to in vitro studies, we also conducted in vivo studies in which cancer patients were given oral supplementation of AJ2 and the function of NK cells were studied. Oral ingestion of AJ2 improved the secretion of IFN-γ by patient derived NK cells and resulted in the better functioning of NK cells in cancer patients. Thus, our studies indicate that for successful NK cell immunotherapy, not only the defect in NK cells but also those in monocytes should be corrected. In this regard, AJ2 probiotic bacteria may serve to provide a potential adjunct treatment strategy.

## 1. Introduction

Natural killer (NK) cells are known as the first line of defense against infections and neoplasia, and they were identified and characterized in the early 1970s [1]. NK cells participate in innate immune function and constitute 10–15% of human peripheral blood lymphocytes [2]. They are identified by their surface expression of CD56 and CD16 and lack of CD3 on their surface. Based on the surface expressions, two distinct populations of NK cells of CD56^dim^CD16^bright^ and CD56^bright^CD16^dim^ were identified having cytotoxicity via perforin–granzyme pathway and immunoregulatory properties via cytokine secretion, respectively [3]. Our previous studies demonstrated that in addition to their cytotoxic function, NK cells have a significant role in promoting differentiation of cancer stem cells (CSCs) by providing critical signals via secreted and membrane bound IFN-γ and TNF-α [4,5,6,7]. IFN-γ produced by NK cells was shown to have great anti-tumor activity due to tumor differentiation [8,9], as well as in an increase expression of CD54 and MHC-class I on tumor cell surfaces [10]. Differentiation of tumors by NK cell-derived IFN-γ was previously shown to directly correlate with the cancer cells’ increased resistance to NK cell-mediated cytotoxicity, their increased sensitivity to chemotherapeutic drugs, and the growth inhibition of tumor cells [10,11].

Monocytes appear to be recruited to tumors and are found to play an important role in cancer progression and metastatic spread of cancer [12,13,14,15]. Human monocytes express CD16, HLADR, CD11b, and CD86, and are classified in three subsets based on CD14 and CD16 surface expression [16,17]. We and other laboratories showed close interactions between NK cells and monocytes, especially their role in the recruitment and activation of NK cells within the tumor microenvironment (TME) [18,19,20,21]. Monocyte-derived IL-15 signaling was found to be required for cytotoxic NK cell-recruitment to the tumor sites [18], and the preliminary activation of monocytes was essential for NK cell proliferation [22]. We also previously demonstrated that monocytes synergize with NK cells in the presence of a combination of eight strains of sonicated probiotic bacteria, sAJ2, to induce CSCs differentiation [5]. Depending on whether monocytes will interact with NK cells first or directly with the tumors, the outcome could be completely different. In the former by activating NK cells monocytes can aid in elimination of the tumor, however, if they interact directly with the tumors, monocytes will deliver survival signals to the tumors providing protection for the tumors [23].

In our previous study, we demonstrated that NK cells’ function, and expansion were suppressed in mice at preneoplastic and neoplastic stages as well as in cancer patients (Appendix A) [24,25,26,27]. Cancer patients exhibit lower NK cell proliferation and demonstrate decreased production of IFN-γ and TNF-α (Appendix A) [27,28,29]. NK cells express several important activating and inhibitory surface receptors, including CD16, and the balance between activating and inhibitory signals which NK cells receive through their surface receptors determines NK cells’ functional fate [30,31,32]. CD16 receptor plays a significant role not only in cytotoxicity and increased secretion of IFN-γ by the NK cells but also in mediating antibody-dependent cellular cytotoxicity (ADCC) by NK cells [33,34,35,36]. In cancer patients, impairment of NK cells’ CD16 surface expression and function was demonstrated [37,38,39]. Studies also showed that NK cells associated with TME are unresponsive to CD16 receptor stimulation, resulting in diminished NK cell-mediated cytotoxicity against tumors [40]. In addition, NK cell mediated secretion of IFN-γ and TNF-α as well as their percentages were found to be impaired in association with tumor-associated monocytes/macrophages [40].

In this study, we sought to focus on identifying the mechanisms responsible for the lack of IFN-γ secretion by the patient derived NK cells, which is responsible for the inability to drive the differentiation of cancer stem cells. In this regard, we studied the role of autologous and allogeneic monocytes from healthy individuals and those of the cancer patients in activating the function of NK cells; in particular, activation of IFN-γ secretion by the NK cells and the functional capability of secreted IFN-γ in driving differentiation of the tumor cells. We compared the role of CD16 signaling as well as bacteria-mediated induction and function of IFN-γ-mediated differentiation of the tumor cells. Although the majority of patients recruited to the study had pancreatic cancer, we also selected to include cancer patients with other types of cancers since, in previous studies, most if not all cancer patients were shown to have defective NK function [41,42,43]. 

## 2. Materials and Methods

### 2.1. Cell Lines, Reagents, and Antibodies

Oral squamous carcinoma stem cells (OSCSCs) were isolated from patients with tongue tumors at the University of California, Los Angeles (UCLA) [11] and were cultured in RPMI 1640 (Life Technologies, Carlsbad, CA, USA) supplemented with 10% fetal bovine serum (FBS) (Gemini Bio-Product, CA, USA). RPMI 1640 supplemented with 10% FBS was used to culture human NK cells and monocytes. Anti-CD16 mAbs, and flow cytometric antibodies were purchased from Biolegend (San Diego, CA, USA). Recombinant human IL-2 was obtained from Hoffman (La Roche, NJ, USA). AJ2 is a combination of seven different strains of gram-positive probiotic bacteria: *Streptococcus thermophiles*, *Bifidobacterium longum*, *Bifidobacterium breve*, *Bifidobacterium infantis*, *Lactobacillus acidophilus*, *Lactobacillus plantarum*, and *Lactobacillus casei*. AJ2 were sonicated as described previously [5]. RPMI 1640 supplemented with 10% FBS was used to re-suspend sonicated AJ2 (sAJ2). Human ELISA kits for IFN-γ were purchased from Biolegend (San Diego, CA, USA). Phosphate buffered saline (PBS) and bovine serum albumin (BSA) were purchased from Life Technologies (Carlsbad, CA, USA). 

### 2.2. Purification of Human NK Cells and Monocytes

Written informed consents approved by the UCLA Institutional Review Board (IRB) were obtained from healthy donors and cancer patients. The study was conducted according to the guidelines of the Declaration of Helsinki and approved by the Institutional Review Board of the University of California, Los Angeles (#11-000781, expiration date 1 December 2021). We used in total 40 cancer patients with 10 different types of cancer, and all cancer patients were at stage 4 (Appendix A). Peripheral blood mononuclear cells (PBMCs) were isolated from peripheral blood as previously described [44]. Briefly, PBMCs were obtained after Ficoll-hypaque centrifugation and were used to isolate NK cells and monocytes using the EasySep^®^ Human NK cell and EasySep^®^ Human Monocytes enrichment kits, respectively, purchased from stem cell technologies (Vancouver, BC, Canada). Isolated NK cells and monocytes were stained with anti-CD16 and anti-CD14 antibodies, respectively, to measure the cell purity using flow cytometric analysis.

### 2.3. Enzyme-Linked Immunosorbent Assays (ELISAs) and Multiplex Cytokine Assay

Single ELISAs were performed as previously described [44]. To analyze and obtain the cytokine and chemokine concentration, a standard curve was generated by either two- or three-fold dilutions of recombinant cytokines provided by the manufacturer. For multiple cytokine array, the levels of cytokines and chemokines were examined by multiplex assay, which was conducted as described in the manufacturer’s protocol for each specified kit. Analysis was performed using a Luminex multiplex instrument (MAGPIX, Millipore, Billerica, MA, USA), and data were analyzed using the proprietary software (xPONENT 4.2, Millipore, Billerica, MA, USA).

### 2.4. ^51^Cr Release Cytotoxicity Assay

The ^51^Cr release cytotoxicity assay was performed as previously described [45]. Briefly, different numbers of effector cells were incubated with ^51^Cr–labeled target cells. After a 4 h incubation period, the supernatants were harvested from each sample, and the released radioactivity was counted using the gamma counter. The percentage specific cytotoxicity was calculated as follows: (1)$$\%\mathrm{cytotoxicity}=\frac{\mathrm{Experimental}\;\mathrm{cpm}-\mathrm{spontaneous}\;\mathrm{cpm}}{\mathrm{Total}\;\mathrm{cpm}-\mathrm{spontaneous}\;\mathrm{cpm}}$$

Lytic units (LU) 30/10^6^ is calculated by using the inverse of the number of effector cells needed to lyse 30% of tumor target cells ×100. 

### 2.5. Enzyme-Linked Immunospot (ELISpot) Assay

The ELISpot were conducted according to manufacturer’s instructions. Briefly, the plate was coated with primary antibody overnight at 4 °C. After washing the plate with PBS twice, desirable cell number were added into each well (40,000 cells/well for PBMCs, NK cells, and NK+monocyte coculture experiments) and incubate at 37 °C for 16–18 h. The plate was washed with PBS and wash buffer twice after the incubation period and detection antibody was added into each well and incubated at room temperature for 2 h. After the incubation period, the plate was washed three times with wash buffer (0.05% Tween20/PBS). Tertiary solution was added into each well and the plate was incubated at room temperature in dark for 30 min. The plate was washed twice with wash buffer and twice with DI water before the blue development solution was added into each well and was incubated for 15 min in dark at room temperature. The reaction was stopped by gently rinsing the plate with water three times, and the plate was then air-dried for 24 h before being read. The number of IFN-γ secreting cells was analyzed using Human IFN-γ Single-Color Enzymatic ELISPOT Assay, ImmunoSpot^®^ S6 UNIVERSAL analyzer and ImmunoSpot^®^ SOFTWARE (all CTL Europe GmbH, Bohn, Germany).

### 2.6. Differentiation of OSCSCs Tumors 

Human NK cells and monocytes were purified from healthy individuals’ and cancer patients’ PBMCs as described above. NK cells and monocytes were treated with IL-2 (1000 U/mL) alone or with a combination of IL-2 (1000 U/mL) and anti-CD16 mAb (3 µg/mL) or with a combination of IL-2 (1000 U/mL) and sAJ2 (NK cells:sAJ2, 1:2). A crisscross NK cells and monocyte cocultures were performed. After 18 h of coculture, supernatants were harvested. Differentiation of OSCSCs was conducted by adding 100 µL supernatant on day 0 and day 1, respectively. On day 3, tumor cells were rinsed with 1X PBS, detached, and used to detect CD54 and MHC-class I on their surfaces using flow cytometer.

### 2.7. AJ2 Dosage to Cancer Patients

Cancer patients were given oral supplementation of 125 billion CFU/capsule, and in total, three capsules/day for four weeks.

### 2.8. Surface Staining Assay

For surface staining, the cells were washed twice using ice-cold PBS + 1%BSA. Predetermined optimal concentrations of specific human monoclonal antibodies were added to 1 × 10^5^ cells in 50 µL of cold PBS + 1%BSA and were incubated on ice for 30 min. Thereafter, cells were washed in cold PBS + 1%BSA and brought to 500 µL with PBS + 1%BSA. Flow cytometric analysis was performed using the Beckman Coulter Epics XL cytometer (Brea, CA, USA), and the results were analyzed in the FlowJo vX software (Ashland, OR, USA). 

### 2.9. Statistical Analyses

All statistical analyses were performed using the GraphPad Prism-8 software. An unpaired or paired, two-tailed Student’s *t*-test was performed for experiments with two groups. One-way ANOVA with a Tukey posthoc test was used to compare different groups for experiments with more than two groups. (n) denotes the number of human donors or number of samples for each experimental condition. Duplicate or triplicate samples were used in the in vitro studies for assessment. The following symbols represent the levels of statistical significance within each analysis: **** (*p* value < 0.0001), *** (*p* value < 0.001), ** (*p* value 0.001–0.01), * (*p* value 0.01–0.05).

## 3. Results

### 3.1. Significantly Decreased CD19+ and Increased CD14+ Cells in Cancer Patients’ PBMCs; Decreased IFN-γ, GM-CSF, IL-1β, IL-7, IL-12, and IL-13 Secretion in Cancer Patients’ Peripheral-Blood Derived Sera

To evaluate the proportions of immune cell subsets in peripheral blood of cancer patients and those of the healthy individuals, we performed flow cytometric analysis using peripheral blood-derived mononuclear cells (PBMCs). Slightly increased percentages of CD16+ NK cells and decreased percentages of CD3+ T cells were found in cancer patients’ PBMCs although there was no statistical significance (Appendix A). Significantly decreased percentages of CD19+ B cells and increased percentages of CD14+ monocytes were found in cancer patients’ PBMCs (Appendix A). Moreover, decreased levels of IFN-γ, GM-CSF, IL-1β, IL-7, IL-12, and IL-13 secretion were seen in cancer patients’ peripheral-blood derived sera when compared to those from the healthy individuals (Appendix A).

### 3.2. Suppressed NK Cell-Mediated Cytotoxicity and Secretion of IFN-γ by Cancer Patients’ PBMCs 

We next assessed the NK cell-mediated cytotoxicity against oral squamous carcinoma stem cells (OSCSCs), and IFN-γ secretion from PBMCs obtained from cancer patients and healthy individuals. PBMCs were left untreated, treated with IL-2, or with the combination of IL-2 and anti-CD16 mAbs, or with the combination of IL-2 and anti-CD3/28 mAbs, or with the combination of IL-2 and probiotic bacteria sAJ2 before they were used in 4 h chromium release assay (Figure 1A,D), or in ELISpot (Figure 1B,E), or in ELISA (Figure 1C,F). Cancer patients’ PBMCs mediated significantly lower levels of cytotoxicity against OSCSCs (Figure 1A,D), and secreted significantly lower amounts of IFN-γ (Figure 1B,C,E,F). These findings indicated that cancer patients’ PBMCs exhibited substantially lower cytotoxicity against cancer stem cells (CSCs) and decreased secretion of IFN-γ in comparison to those from healthy individuals. Due to significant variability in the results from day-to-day experiments, we selected to present the results from patient and age/sex matched healthy donors as the representative experiment since they were run on the same day using the same reagents, which is more representative of differences observed (Figure 1A–C). We also compiled all the patients and healthy donor results in the scatter plot, even though the variability is likely to mask the significant differences which were seen between the patients and the healthy donors (Figure 1D–F). 

### 3.3. Cancer Patients’ Monocytes Suppressed the Cytotoxic Activity of Both Autologous and Allogeneic Healthy NK Cells, whereas Healthy Individuals’ Monocytes Increased Cytotoxic Activity in NK Cells 

To determine the functional interaction between NK cells and monocytes from cancer patients and those from the healthy individuals, we cocultured autologous and allogeneic NK and monocytes derived from cancer patients and healthy individuals. In the absence of monocytes, NK cells from cancer patients exhibited decreased cytotoxic function when treated with IL-2 alone (Figure 2A and Appendix A), or IL-2+anti-CD16 mAbs (Figure 2B and Appendix A), or IL-2 + sAJ2 (Figure 2C and Appendix A). In the presence of cancer patients’ monocytes, the cytotoxic function was suppressed both in autologous and allogeneic healthy NK cells when they were treated with IL-2 alone (Figure 2A and Appendix A), or with IL-2 + anti-CD16 mAbs (Figure 2B and Appendix A), or with IL-2 + sAJ2 (Figure 2C and Appendix A) when compared to that of NK cells in the absence of monocytes. 

In the presence of healthy individuals’ monocytes, the cytotoxic function was increased in autologous healthy NK cells when treated with IL-2 alone (Figure 2A and Appendix A), or IL-2 + anti-CD16 mAbs (Figure 2B and Appendix A) or IL-2 + sAJ2 (Figure 2C and Appendix A). However, when healthy monocytes were cocultured with cancer patients’ NK cells, the cytotoxic function of NK cells was slightly suppressed when treated with IL-2 alone (Figure 2A and Appendix A) and was increased when treated with IL-2 + anti-CD16 mAbs (Figure 2B and Appendix A) or IL-2 + sAJ2 (Figure 2C and Appendix A) when compared to that of patient NK cells in the absence of monocytes. Overall, these results indicated that cancer patients’ monocytes failed to increase NK cell cytotoxicity both in patient and healthy NK cells. 

### 3.4. Cancer Patients’ Monocytes in Comparison to Healthy Individuals’ Monocytes Induced Lower Increase in IFN-γ Secretion Both in Autologous and Allogeneic NK Cells

To assess cancer patients’ and healthy individuals’ monocyte-induced effect on NK cell-mediated increase in IFN-γ secretion, we cocultured autologous and allogeneic NK and monocytes and determined the number of IFN-γ spots using ELISpot assay (Figure 3 and Appendix A), and secretion using ELISA (Figure 4). In the absence of monocytes, NK cells from cancer patients exhibited suppressed IFN-γ secretion when treated with IL-2 alone (Figure 3A,B, Figure 4A and Appendix A), or IL-2 + anti-CD16 mAbs (Figure 3A,C, Figure 4B and Appendix A), but increased IFN-γ secretion in cancer patients’ NK cells were seen when treated with IL-2 + sAJ2 (Figure 3A,D, Figure 4C and Appendix A). Although the presence of both cancer patients’ and healthy individuals’ monocytes increased IFN-γ secretion, the higher increase in IFN-γ secretion was seen by healthy individuals’ monocytes cultured with both autologous and allogenic NK cells in the presence of IL-2 + anti-CD16 mAbs treatment when compared to that of IL-2 treatment alone. (Figure 3A–C and Figure 4A,B). Treatment with IL-2 + sAJ2 increased the IFN-γ secretion in both patients and healthy individuals’ NK cells. Likewise, increased levels of IFN-γ were seen when IL-2+sAJ2 treated NK cells were cultured with either cancer patients’ or healthy individuals’ monocytes (Figure 3A,D and Figure 4C). Lower IFN-γ secretion was seen in patient NK cells alone or in the presence of monocytes when secretion levels were determined using multiplex luminex assay (Appendix A). Collectively, these findings indicated that cancer patients’ monocytes have lower NK activating capacity to secrete IFN-γ when compared to those from healthy individuals.

### 3.5. Cancer Patients’ NK Cells when Cocultured with Healthy Monocytes and Treated with sAJ2 Induced Higher IFN-γ and Elevated Surface Receptor Expressions Associated with the Differentiation of Tumor Cells

As previously described, NK cells play important roles in differentiating CSCs, leading to slower tumor growth, and decreased tumor metastasis [26,46]. Therefore, we cocultured autologous and allogeneic NK and monocytes and used their supernatants to differentiate OSCSCs. The surface expressions of CD54 and MHC-class I were determined on the surface of oral tumors three days after treatment with the supernatants. We observed higher surface expression of CD54 (Figure 5A and Appendix A), and MHC-class I (Figure 5B and Appendix A), when cocultures were treated with IL-2 + sAJ2 in comparison to IL-2 alone or IL-2 + anti-CD16 mAbs. Also, higher surface expressions of CD54 (Figure 5A and Appendix A), and MHC-class I (Figure 5B and Appendix A) on tumor cells were found when supernatants from both healthy and patient NK cells cultured with healthy monocytes were used to differentiate tumors in comparison to using patient monocytes. Next, we determined the ratios of the density of surface expressions for CD54 and MHC class I (as represented by MFI) in anti-CD16 mAbs or sAJ2 treated NK cells in the presence and absence of autologous and allogeneic monocyte cocultures treated with and without sAJ2 (Figure 5C,D). The fold change in the expression levels between NK cells alone and those cultured with the monocytes when autologous patient NK cells are used in the cocultures after activation with IL-2 and anti-CD16 mAbs (1.16-fold for CD54 expression and 1.2-fold for MHC-class I expression) was much less when compared to that of NK cells and autologous monocyte cocultures from healthy individuals (1.74-fold increase for CD54 and 1.8 for MHC-class I) (Figure 5C,D). Addition of sAJ2 to cocultures highly improved the fold change between NK cells alone and those cultured with monocytes between patients and healthy individuals (1.7-fold change for CD54 and 2.2-fold change for MHC-class I expression in patient cocultures vs. 2.0-fold change for those from healthy individuals for CD54 expression, and 1.55-fold change for MHC-class I for those from healthy individuals) (Figure 5C,D). The addition of monocytes from healthy individuals to patient NK cells increased the levels of IFN-γ secretion and augmented the levels of differentiation in OSCSCs (Figure 3, Figure 4 and Figure 5). Supernatants from patient monocytes cultured with healthy NK cells demonstrated decreased ability to differentiate tumor cells, when compared to those from healthy NK cells cultured with autologous monocytes; however, the levels of tumor differentiation were higher when compared to autologous patient NK cocultures with monocytes (Figure 5). Therefore, either the addition of monocytes from the healthy individuals to patient NK cells or treatment of patient NK cells with autologous monocytes with sAJ2 improved patient NK cell-mediated differentiation of tumor cells substantially.

### 3.6. Decreased CD16 Surface Expression on Monocytes Obtained from Cancer Patients

To further understand the mechanism contributing to decreased monocyte-induced NK cell activation, we next determined the levels of CD16 surface expression on CD14+ monocytes of healthy individuals and those of the cancer patients using flow cytometry. There was a slight decrease in the percentages of CD16-positive monocytes in cancer patients (Figure 6A,B); however, even though all monocytes from cancer patients exhibited some CD16 membrane expression, the intensity and levels of expression were significantly lower in patient monocytes when compared to those from healthy individuals, as evidenced by significantly decreased mean channel fluorescence (MFI) (Figure 6A,C). We also determined HLADR, CD33, and CD11b surface expression on CD14+ monocytes and found lower HLADR, higher CD33, and similar levels of CD11b surface expressions on patient monocytes in comparison to those of healthy monocytes (Appendix A). Thus, lower expression of CD16 and HLADR surface receptors were observed on patients’ monocytes when compared to those from healthy individuals.

### 3.7. Increased IFN-γ Secretion, and NK Cell-Mediated Cytotoxicity in Cancer Patients’ Peripheral Blood-Derived NK Cells in Response to AJ2 Oral ingestion

After four weeks of oral ingestion of AJ2 probiotic bacteria (125 billion CFU/capsule: three capsules per day) NK cells from pancreatic cancer patients demonstrated increased IFN-γ secretion (Figure 7A and Appendix A), and NK cell-mediated cytotoxicity (Figure 7B and Appendix A) in peripheral blood-derived NK cells. Significantly decreased NK cell function was observed before AJ2 ingestion in cancer patients (Appendix A). Indeed, the fold difference in the NK-mediated IFN-γ secretion between healthy and patient before oral supplementation was very high at 2.9-fold, whereas after supplementation the difference became much less at 1.2-fold (Figure 7A). The fold difference in the NK cell-mediated cytotoxicity between healthy and patient before oral supplementation was 2.7-fold, whereas after supplementation it was 1.4-fold (Figure 7B). Therefore, AJ2 resulted in improvement of IFN-γ secretion and cytotoxicity by the NK cells in cancer patients.

## 4. Discussion

Previous studies from our laboratory and those of the others demonstrated defects in NK cell function in cancer patients (Appendix A); however, the underlying mechanisms for such defects was not clearly delineated, nor is it known whether other immune effectors such as monocytes that are known to activate NK cells are also defective in their function in cancer patients [7,47]. Therefore, in this paper, we show that NK cell-mediated cytotoxicity and induction of IFN-γ in PBMCs of cancer patients are significantly defective when compared to those obtained from healthy donors. Although the defect can be seen in most in-vitro treatments tested including those triggered by IL-2 or the combinations of IL-2 with anti-CD16 mAbs, or IL-2 with anti-CD3/28 mAbs or IL-2 with sAJ2 in PBMCs, the most significant defect was seen in those treated with IL-2 and anti-CD16 mAbs, and the least in those treated with IL-2 and sAJ2. Since we previously observed decreased expression of CD16 on patient NK cells [46] and significant differences in the function of PBMCs when they were activated with IL-2 and anti-CD16 mAbs, we undertook studies to understand the underlying mechanisms of the insufficient NK cell activation by anti-CD16 mAbs. We purified peripheral blood NK cells and studied their functions following cocultures with autologous and allogeneic monocytes obtained from cancer patient and those of the healthy individuals in the presence of IL-2 and anti-CD16 mAbs activation and compared the effect to those activated with IL-2 and sAJ2. As presented in the Section 3, the levels of cytotoxicity were decreased when cancer patients’ or healthy individuals’ NK cells were cultured in the presence of patient monocytes, as compared to those of healthy monocytes in a representative experiment. However, when we compiled the data from different donors, the differences were less dramatic, but the patterns remained consistent in the presence of IL-2 treatment. This is likely due to the large variability we see in the values between the donors and due to experimental procedures performed on different days. Interestingly, treatment with sAJ2, unlike anti-CD16 mAbs, maintained higher NK cell-mediated cytotoxicity in the patient and healthy NK cells cultures with both patient and healthy monocytes. Healthy and patient NK cells in the presence of healthy monocytes generally had higher cytotoxicity, when compared to those cultured with patient monocytes and treated with IL-2 or IL-2 with anti-CD16 mAbs. However, the experimental conditions were optimized to observe the differences between NK cultures with monocytes and not NK cells alone. Treatment with anti-CD16 mAbs induced split anergy in NK cells, leading to decreased cytotoxicity in the presence of increased IFN-γ secretion [48]. Furthermore, sAJ2 treatment maintained or increased cytotoxicity, especially in patients’ NK cells with autologous and allogeneic monocytes. This could be due to internalization or shedding of CD16 receptors, as compared to toll-like receptors (TLR). We previously hypothesized and showed that decreases in CD16 receptors and inhibition of NK cell-mediated cytotoxicity could be a physiological programming for NK cells to switch their phenotype from CD16^bright/+^ to CD16^low/−^ to increase secretion of IFN-γ and TNF-α, while decreasing their cytotoxicity to differentiate tumor cells that were selected by the NK cells [4]. Such physiological conditioning of NK cells is not only important for defense against infections, trauma, and other injuries, but also against tumor cells. By increasing differentiation of tumors, this change in the phenotype of the NK cells can lead to growth inhibition of tumor cells and lower rate of tumor expansion, while increasing tumor susceptibility to a number of other therapeutic strategies, such as chemotherapy, radiotherapy, effect by checkpoint inhibitors, and T cell-mediated effects. The downside of such NK conditioning is the potential survival of some differentiated tumors since these tumors are not targeted or killed by the primary NK cells; however, such tumors should be killed by CD8+ T cells, since they exhibit higher levels of MHC-class I surface expression. 

When cultured with monocytes, NK cells also either decrease, maintain the same level, or in very few cases slightly increase the lysis of tumor cells [7]; however, they always substantially increase the levels of induction and secretion of IFN-γ. Indeed, the only time an increase in NK cell-mediated cytotoxicity by monocytes was seen was when the cocultures were treated with sAJ2; however, the differences were not statistically significant. Therefore, monocytes can also be regarded as important effectors in inducing split anergy in NK cells as a mechanism to drive differentiation of tumor cells.

Our results demonstrate that patient monocytes induced lower levels of IFN-γ spots, as well as IFN-γ secretion, when cultured with autologous or allogeneic NK cells in ELISpot and ELISA assays, respectively. Monocytes from healthy individuals induced higher levels of IFN-γ spots in cultures with both autologous and allogeneic NK cells from patients. Therefore, monocytes obtained from healthy individuals were capable of increasing and restoring the IFN-γ induction of patient NK cells in IL-2 or IL-2 and anti-CD16 mAbs-treated groups. Thus, infusing allogeneic monocytes from healthy individuals to cancer patients could be another strategy to increase patient NK cell function. Interestingly, treatment with sAJ2 was also able to increase and restore IFN-γ induction in patient NK cells cultured with autologous monocytes. On average, the differences between patient NK cells cultured with either healthy or patient monocytes and healthy NK cells cultured with patient and healthy monocytes when treated with sAJ2 as opposed to anti-CD16 mAbs were very slight when assessed using the ELISpot assay. In agreement with the much-improved results seen in patient NK cells when they were treated with sAJ2 in the presence of monocytes, patients after ingesting AJ2 probiotic bacteria exhibited improved NK cell-mediated cytotoxicity and secretion of IFN-γ when compared to those from healthy individuals (Figure 7 and Appendix A). Whether such improved profiles of NK cell-mediated cytotoxicity and secretion of IFN-γ in patients have long lasting effect on disease reversal or delay in disease progression requires further studies. 

We previously established that differentiated OSCSC tumors express higher levels of CD54, MHC-class I, and B7H1, but lower levels of CD44 (Appendix A) [48]. We also showed that equal amounts of IFN-γ secreted from patients’ NK cells as opposed to those from healthy individuals have much lower ability to increase differentiation of tumor cells demonstrating defect in the function of secreted IFN-γ by patients’ NK cells [29]. Thus, we sought to determine the functional ability of IFN-γ produced by patient NK cells in comparison to those obtained from healthy individuals in the presence and absence of autologous and allogeneic monocytes. Similar to the profiles obtained by ELISpot and ELISA, we saw decreased levels of induction of CD54 and MHC-class I on tumor cells by supernatants of patients as compared to those from the healthy individuals when the NK cells were treated with the combination of IL-2 and anti-CD16 mAbs. As mentioned above, CD54 and MHC-class I are the markers of tumor differentiation that we established previously in a great number of papers (Appendix A) to correlate with the NK cell-mediated differentiation of tumors. Indeed, the fold change in the expression levels between NK cells alone and those cultured with the monocytes when autologous patient NK cells are used in the cocultures after activation with IL-2 and anti-CD16 mAbs was much less when compared to NK cells and autologous monocyte cocultures from healthy individuals. Addition of sAJ2 to cocultures highly improved the fold change between NK cells alone and those cultured with monocytes between patients and healthy individuals. The addition of monocytes from healthy individuals to patient NK cells increased the levels of IFN-γ secretion and augmented the levels of differentiation in OSCSCs. Supernatants from patient monocytes cultures with healthy NK cells demonstrated decreased ability to differentiate tumor cells when compared to those from healthy NK cells cultured with autologous monocytes; however, the levels of tumor differentiation were higher when compared to autologous patient NK cocultures with monocytes. Therefore, either the addition of monocytes from the healthy individuals to patient NK cells or supplementation of patient NK cells with autologous monocytes with sAJ2 improved patient NK cell-mediated differentiation of tumor cells substantially. The effect of sAJ2 on the cultures of NK cells with monocytes is different from those of CD16-mediated effects in patients, since it appears that treatment with anti-CD16 mAbs either does not improve or improves only moderately the differentiation markers of tumors by IFN-γ from the patients’ NK/monocyte cocultures, when compared to those from healthy individuals. At present it is not clear whether the ineffectiveness of CD16-mediated effect in patient NK/monocyte cocultures is unique to CD16 or other key NK/monocyte receptors are equally defective in their function. Therefore, the decreased capacity of NK cells to differentiate tumors should be regarded as one of the major causes of survival and expansion of poorly differentiated tumors in cancer patients, since lack of adequate differentiation of the tumors will allow the survival and expansion of poorly differentiated tumors and promotion of metastasis. Indeed, it is possible that what is known by dormant tumor niches in different organs are the consequences of immune cell function such as NK cells in increased differentiation of these tumors [49]. Thus, treatment with sAJ2 may partly serve to restore the loss in ability of patient NK cell derived IFN-γ to differentiate tumors. In addition, higher increase in MHC-class I on tumor cells with the supernatants from the combination of sAJ2 treated NK-monocyte cocultures not only decreases the proliferation and expansion of differentiated tumors, but also these tumors are likely to regulate the function of NK cells by inhibiting additional signaling, potentially leading to cessation of inflammation, which allows NK cells to recover from over activation. Whether the secreted IFN-γ in patients is complexed by its shed receptors or that IFN-γ is bound to an inhibitor to prevent their differentiation function in patients requires future investigation. 

We next determined the levels of CD16 receptors on monocytes and found that CD16 receptor expression was significantly decreased on the surface of patient monocytes, when compared to those of healthy individuals. Downmodulation and/or shedding of CD16 receptors from NK cells [46] compounded with the same effects in monocytes may present one of the underlying mechanisms of dysfunction in NK cells. Currently, we are in the process of delineating the role of such downmodulation in the function s of NK cells.

Ingestion of AJ2 by patients demonstrated improvement in the NK function both in terms of IFN-γ secretion as well as NK cell-mediated cytotoxicity after four weeks of consumption. Increased function of NK cells seen both in in vitro and in vivo studies points to the significance of bacteria-mediated increase in immune activation, and restoration of immune function in patients. Whether longer consumption of probiotic bacteria will maintain increased NK function requires future studies. In addition, we showed in vivo previously in the humanized-BLT mouse model that the ingestion of AJ2 probiotic bacteria increased NK cell function and correlated with the decrease or elimination of oral and pancreatic tumors [25,26]. 

We previously found that monocytes are one of the major cell types in imparting resistance to cell death in tumor cells [50]. This observation is of utmost significance, since depending on which cell type monocytes initially interact with, i.e., immune cells or the tumor cells, the fate of the tumor cells may be different. If they interact with the competent NK and T cells, it is likely monocytes will increase the functional capabilities of immune cells to target the tumor cells. However, if monocytes interact with tumor cells, they may elevate tumor resistance to NK and T cell-mediated cell death. Therefore, elevation in monocyte expansion in the absence of NK or T cell expansion and function in cancer patients may be detrimental since there may be a greater chance of monocytes to interact with tumor cells than the immune cells, which will likely cause the resistance and survival of tumor cells. Indeed, NK cells are not only activated by monocytes, but during such interaction they also eliminate these cells. Therefore, it is imperative to have great numbers of functionally competent NK cells to bind to and interact with monocytes, since such NK cells will not only eliminate the cancer stem cells/poorly differentiated tumors, but also eliminate monocytes and do not allow them to interact with tumors directly within the tumor microenvironment. If the number of monocytes rise in patients in the presence of decreased numbers of functionally competent NK and T cells, this could be a troubling sign since monocytes may end up aiding the tumors to survive instead of helping NK or CD8+ T cells to increase their respective functions. Indeed, we observed such profiles in terminally ill cancer patients at the later stages of cancer progression [51]. Therefore, the roles of NK cells are several fold within the tumor microenvironment. Such complex interactions within the tumor microenvironment and peripheral blood of patients are the focus of our future studies to predict clinical outcomes in cancer patients.

## Figures and Tables

**Figure 1 cells-11-00697-f001:**
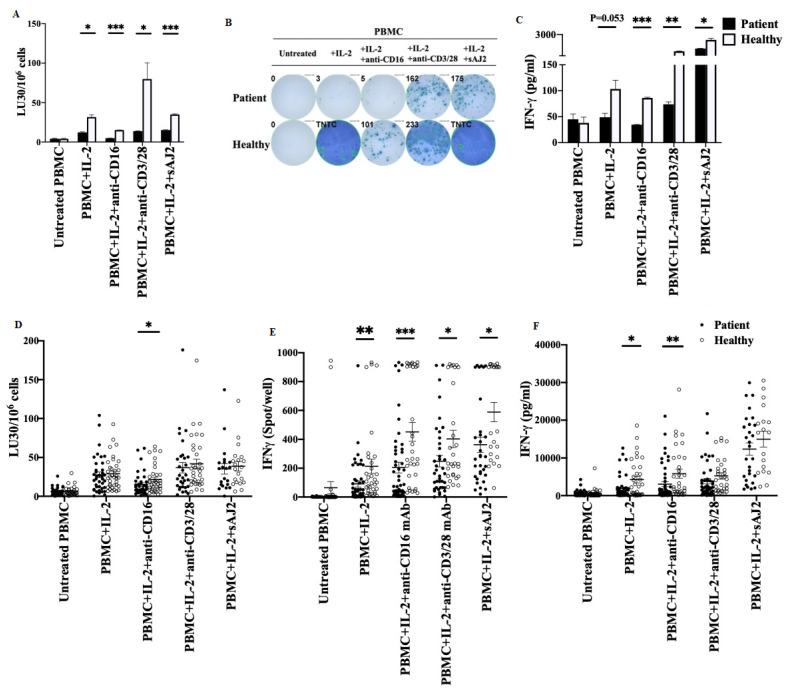
Peripheral blood mononuclear cells obtained from cancer patients exhibited lower NK cell-mediated cytotoxicity and IFN-γ secretion. PBMCs were isolated from cancer patients’ and those of the healthy individuals’ peripheral blood as described in Section 2. PBMCs were left untreated or treated with IL-2 (1000 U/mL) or with a combination of IL-2 (1000 U/mL) and anti-CD16 mAbs (3 µg/mL), or with a combination of IL-2 (1000 U/mL) and anti-CD3/28 mAbs (25 µL/mL), or with a combination of IL-2 (1000 U/mL) and sAJ2 (PBMC:sAJ2, 1:20) for 18 h before using in functional assays. NK cell-mediated cytotoxicity of PBMCs was determined using standard 4 h ^51^Cr release assay against OSCSCs. Lytic units (LU) 30/10^6^ cells were determined using inverse number of PBMCs required to lyse 30% of OSCSCs × 100 (**A**,**D**). Number of cells secreting IFN-γ in PBMCs were determined as spot counts using ELISpot assay (**B**,**E**). Supernatants were harvested from PBMCs and the secretion of IFN-γ were determined using ELISA (**C**,**F**). A representative experiment is shown in (**A**–**C**) and the data are presented as Mean ± SD. Compiled data are shown in (**D**–**E**) (*n* = 20 to 49), and the data are presented as Mean ± SEM. Student *t* tests were performed. *** (*p* value < 0.001), ** (*p* value 0.001–0.01), * (*p* value 0.01–0.05).

**Figure 2 cells-11-00697-f002:**
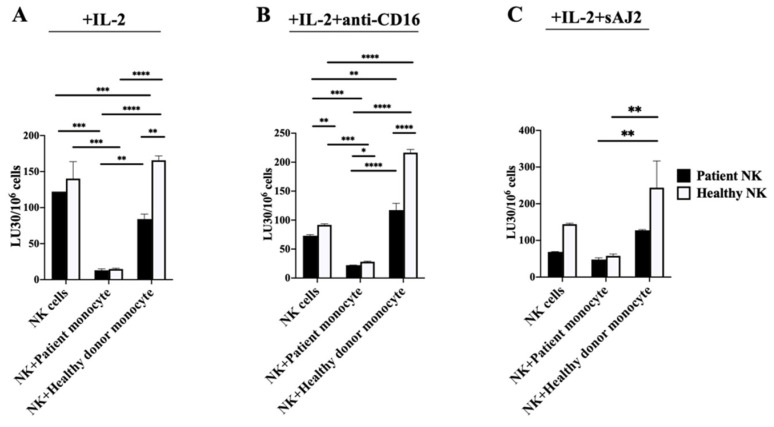
Cancer patients’ monocytes suppressed cytotoxicity in NK cells, whereas increased cytotoxicity in NK cells was seen when they were cultured with healthy individuals’ monocytes. NK cells and monocytes of cancer patients and healthy individuals were isolated from PBMCs as described in Section 2. NK cells and monocytes were treated with IL-2 (1000 U/mL) alone (**A**) or with a combination of IL-2 (1000 U/mL) and anti-CD16 mAbs (3 µg/mL) (**B**), or with a combination of IL-2 (1000 U/mL) and sAJ2 (NK:sAJ2, 1:2) (**C**). Co-cultures of NK cells with autologous and allogeneic monocytes from both the patient and healthy donor were performed. NK cell-mediated cytotoxicity was measured after 18 h of coculture using standard 4 h ^51^Cr release assay against OSCSCs. Lytic units (LU) of 30/10^6^ cells were determined using inverse number of NK cells needed to lyse 30% of target cells OSCSCs × 100. A representative experiment is shown in (**A**–**C**), and data are presented as Mean±SD. Student *t* tests were performed to determine statistical significance. **** (*p* value < 0.0001), *** (*p* value < 0.001), ** (*p* value 0.001–0.01), * (*p* value 0.01–0.05).

**Figure 3 cells-11-00697-f003:**
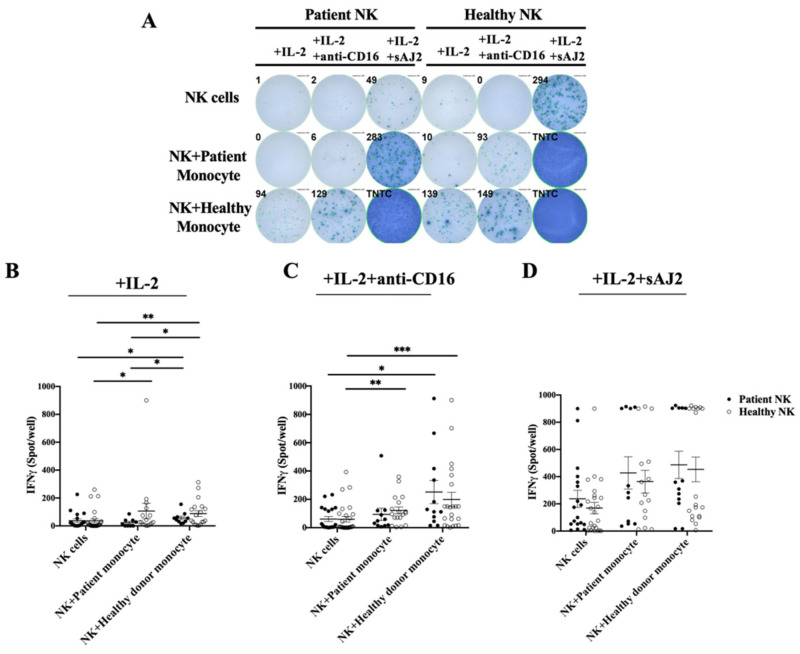
Cancer patients’ monocytes in comparison to healthy individuals’ monocytes are less capable of inducing IFN-γ secretion by autologous and allogeneic healthy NK cells. NK cells and monocytes of cancer patients and those of the healthy individuals were isolated from PBMCs as described in Section 2. NK cells and monocytes were treated with IL-2 (1000 U/mL) alone (**A**,**B**) or with a combination of IL-2 (1000 U/mL) and anti-CD16 mAbs (3 µg/mL) (**A**,**C**) or with a combination of IL-2 (1000 U/mL) and sAJ2 (NK cells:sAJ2, 1:2) (**A**,**D**). Co-cultures of NK cells with autologous and allogeneic monocytes from both the patient and healthy donor were performed. After 18 h of coculture, number of IFN-γ secreting cells were determined using ELISpot assay as spot counts. A representative experiment is shown in (**A**). Compiled data are shown in (**B**–**D**) (*n* = 11 to 28), and data are presented as Mean ± SEM. Student *t* tests were performed to determine statistical significance. *** (*p* value < 0.001), ** (*p* value 0.001–0.01), * (*p* value 0.01–0.05).

**Figure 4 cells-11-00697-f004:**
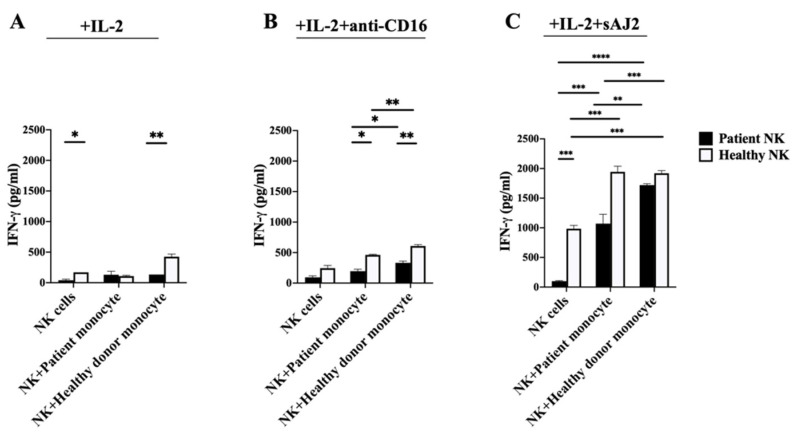
Lower IFN-γ secretion by cancer patients’ NK cells were observed when they were cocultured with autologous or allogeneic healthy monocytes. NK cells and monocytes of cancer patients and those of the healthy individuals were isolated from PBMCs as described in Section 2. NK cells and monocyte were treated with IL-2 (1000 U/mL) alone (**A**), or with a combination of IL-2 (1000 U/mL) and anti-CD16 mAbs (3 µg/mL) (**B**), or with a combination of IL-2 (1000 U/mL) and sAJ2 (NK cells:sAJ2, 1:2) (**C**). Co-cultures of NK cells with autologous and allogeneic monocytes from both the patient and healthy donor were performed. After 18 h of coculture, supernatants were harvested and used in ELISA to measure IFN-γ secretion. A representative experiment is shown, and data are presented as Mean ± SD. Student *t* tests were performed to determine statistical significance. **** (*p* value < 0.0001), *** (*p* value < 0.001), ** (*p* value 0.001–0.01), * (*p* value 0.01–0.05).

**Figure 5 cells-11-00697-f005:**
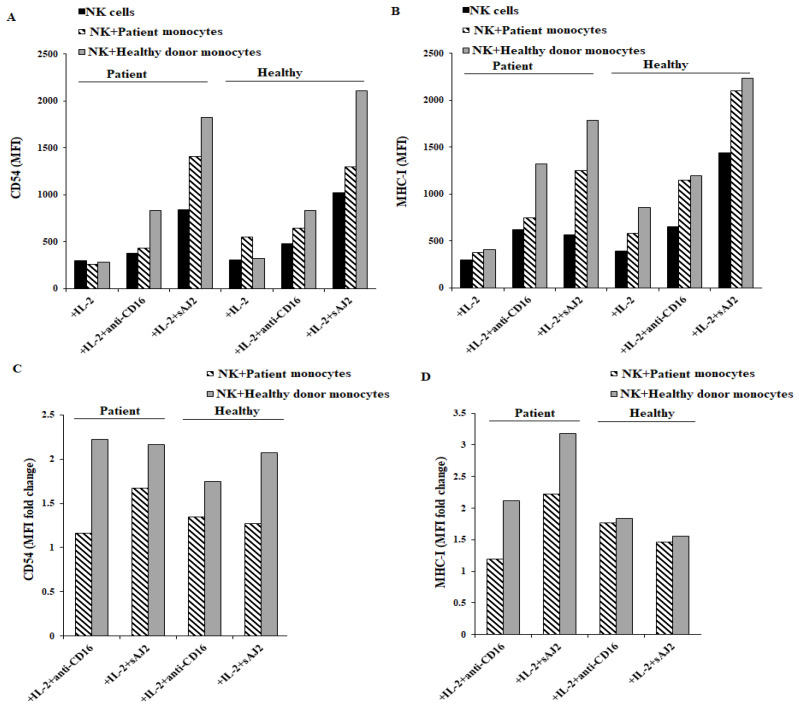
Supernatants harvested from autologous cancer patients’ NK and monocytes coculture treated with IL-2 + sAJ2 induced higher differentiation of oral squamous carcinoma stem cells (OSCSCs) in comparison to those treated with IL-2 alone or IL-2 + anti-CD16 mAbs. OSCSCs differentiation assay were conducted as described in Section 2 using supernatants collected from NK cell and monocyte coculture experiment. Surface expression of CD54 (**A**) and MHC-class I (**B**) on supernatant treated OSCSCs were determined using flow cytometry. MFI are shown in figures (**A**,**B**). MFI ratio of (Nk+monocyte/NK) for CD54 (**C**) and MHC-class I (**D**) surface expression were determined after treatment of tumor cells with supernatants collected from IL-2 + anti-CD16 mAbs or IL-2 + sAJ2-treated cells. One of two representative experiments is shown in this figure.

**Figure 6 cells-11-00697-f006:**
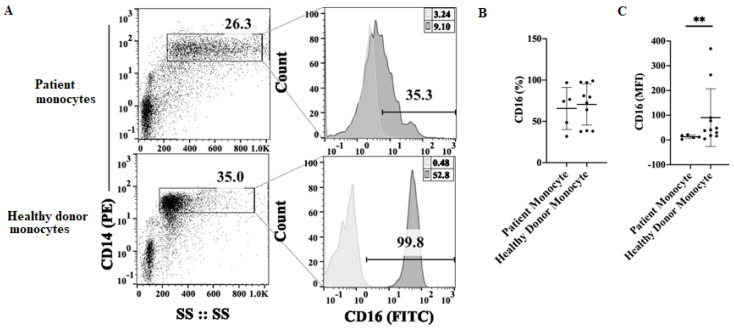
Monocytes obtained from cancer patients exhibited lower CD16 surface receptor expression. Monocytes of cancer patients (*n* = 5) and healthy individuals (*n* = 10) were isolated from PBMCs as described in Section 2. Surface expression of CD16 within CD14+ population was determined using flow cytometry. One representative experiment is shown in (**A**). Compiled experiments are shown in (**B**,**C**), and data are presented as Mean ± SD. Student *t* tests were performed to determine statistical significance. ** (*p* value 0.001–0.01).

**Figure 7 cells-11-00697-f007:**
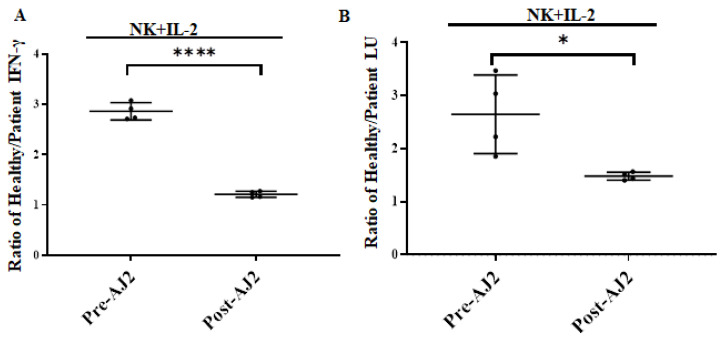
Increased IFN-γ secretion and cytotoxicity in cancer patients’ NK cells after oral AJ2 supplementation. Freshly purified NK cells (1 × 10^6^ cells/mL) from cancer patients and those of the healthy individuals were treated with IL-2 (1000 U/mL) for 18 h before supernatants were harvested to determine IFN-γ secretion using ELISA (Pre-AJ2). Cancer patients were on oral supplementation of AJ2 for four weeks before we analyzed their NK cells. NK cells were treated with IL-2 (1000 U/mL) for 18 h before supernatants were harvested to determine IFN-γ secretion using ELISA (Post-AJ2). Ratio of secreted IFN-γ between healthy and patient NK cells were determined at pre- and post-AJ2 oral supplementation (*n* = 4) (**A**). Freshly purified NK cells (1 × 10^6^ cells/mL) from cancer patients and those of the healthy individuals were treated with IL-2 (1000 U/mL) for 18 h before they were used as effectors in cytotoxicity against OSCSCs using standard 4 h ^51^Cr release assay. LUs were determined as described in Figure 2. (Pre-AJ2). Cancer patients were on oral supplementation of AJ2 for four weeks before analysis of their NK cells. NK cells were treated with IL-2 (1000 U/mL) for 18 h before they were used as effectors in cytotoxicity against OSCSCs using standard 4 h ^51^Cr release assay. LUs were determined as described in Figure 2. (Post-AJ2). Ratio of cytotoxicity between healthy and patient NK cell-mediated cytotoxicity were determined at pre- and post-AJ2 oral supplementation in cancer patients (*n* = 4) (**B**). **** (*p* value < 0.0001), * (*p* value 0.01–0.05).

## Data Availability

The study did not report any data.

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
