# Peer review of "Defective Patient NK Function Is Reversed by AJ2 Probiotic Bacteria or Addition of Allogeneic Healthy Monocytes"

_cells, 2022, doi:10.3390/cells11040697_

Round 1
Reviewer 1 Report
The authors did not respond to the queries. Only minor changes were noted in the manuscript. Taking into account that the group have been working and there are similar results published and assuming this manuscript is a continuation of those previous reports, the manuscript can be accepted. However, the authors must consider that similar data should not be published repetitively. The authors also must include other references not only self-references prior to final approval. For example PMID: 28616355
- PMID: 34835287
Reviewer 2 Report
The authors have satisfactorily addressed the concerns raised in the original version. The revised version is significantly improved. No further concerns.
This manuscript is a resubmission of an earlier submission. The following is a list of the peer review reports and author responses from that submission.
Round 1
Reviewer 1 Report
this is a well written manuscript, could you please add more references from 2019-2021
Reviewer 2 Report
The importance of IFN gamma production in NK cells is crucial in the cytotoxic response of cancer cells. The manuscript which is quite promising by the title requires to be restructured and rewritten in some areas.
Major points
1.- A table concerning patients and control characteristics is lacking and it is crucial. This information is required since it will be related to the monocyte retrieved from the PBMC of these patients. There is also an important question why the authors did not compare their results with K562, the standard cell line of cytotoxicity? Why oral carcinoma?
2.- The second issue is related to monocyte heterogenicity in patients and the expression of CD16, why other cell markers were not used? If there is a crosslink with CD16, why there are no other molecules? Why IL-2 and not IL-15? It is unclear. It is known that anti CD16, 3G8 antibody induces IL-12 which may be crucial in the process.
3.-The analysis of the expression of figure 2 and 3 are misleading since there is a heterogeneity of values. I suggest that the analysis should separate high to low responders so it is easier to envision the changes, ie figures 3C and 3 D
Figure 5 is quite confusing concerning MHC I expression. The authors do not discuss it much. If there is an increase in MHC class I then it is possible to have the resistance to NK cytotoxic activity. This has to be clarified.
Figure 6 part C it involves activated monocytes, high MFI, they should be not used in the calculation of the difference of MFI as compared to patients, essentially CD16 in % does not differ in most of the samples.
The discussion concerning supplying the bacteria to the patients and in vitro activation using AJ2 is also lacking. The use of killed bacteria as stimulation of NK cells have been published and the mechanism differs from the one in vitro. manuscript.
The supplementary files are complete; however, it would be important to add the effect of oral treatment in the patients
The figures of ELISPOT are not very informative and the numbers should be bigger.
The discussion should be rewritten not only based on the data of the group but taking into account other results.
Table 1 is not informative.
The ethical approval for the study is not clearly stated.
There are some sentences in the beginig of results which are not part of the
Reviewer 3 Report
Title: “Defective autologous monocyte induced IFN-γ secretion and function in cancer patient NK cells is reversed by AJ2 probiotic bacteria treatment or addition of allogeniec healthy monocytes”
Authors: Meng-Wei Ko, Kawaljit Kaur, Tahmineh Safaei, Wuyang Chen, Christine Sutanto, Paul Wong, Anahid Jewett
Summary:
The present work addresses the importance of identifying the mechanisms responsible for the lack of IFN-γ secretion by patients' NK cells, which in turn are responsible for the inability to drive cancer stem cell differentiation. The authors examined the role of autologous and allogeneic monocytes from healthy individuals and from cancer patients in activating NK cell function, specifically in the activation of IFN-γ secretion by NK cells and the functional ability of secreted IFN-γ to promote tumor cell differentiation. They compared the role of CD16 signaling and bacterial-mediated induction and function of IFN-γ-mediated tumor cell differentiation.
Major Points:
1: The entire paper needs restructuring and clarification, as the formulations are woolly, the methods one-sided, and the results superficially presented.
2: Furthermore, the text is dominated by self-citations.
3: I have trouble understanding the goal and approach of the study.
Title:
4: The title is too long, too confusing, and contains a spelling error (allogeniec).
Abstract:
5: The abstract is too non-specific for me as it only talks about cancer in general. It is not clear which cancers are being studied and whether it is a vitro or vivo study. Therefore, the abstract is not concise enough and should be rewritten.
Introduction
6: Again, cancer patients and tumors are mentioned, but the type of cancer and the relevance of the study are not clarified.
Materials/Methods
7: 2.1. Write out/explain UCLA, since not everyone knows the institute.
8: 2.2. Four types of cancer are mentioned here. Are the cells studied individually and compared or are they mixed?
9: 2.6. It is now said that the NK cells are mixed. Why is this done instead of focusing on one type of cancer?
Results
10: The introduction could be worded more elegantly, it sounds more like a work order.
11: 3.1: Cytokines and chemokines is too non-specific for me. Which ones are meant?
12: 3.2: OSCSCs suddenly come into play here. This makes it even more unclear to me which clinical target the study is aiming at.
13: 3.7.: Only NK cells in pancreatic cancer are mentioned here (instead of the 4 cancer types in M&M). Why?
14: Overall, I am confused by the variety of different cancer types. This makes it difficult for me to understand the aim and outcome of the study. I find the methods biased and miss e.g., WB and real photos.
Discussion
15: Also, in the discussion the "results" are somewhat lost, since much is written around them and repeated (suggestion for improvement: scheme or logical listing of the key statements.
The last paragraph is about terminally ill cancer patients. However, the cancer stages of the studied subjects were not mentioned.
Tables
16: Both tables are not meaningful enough and too inaccurate. What types of cancer are we talking about here? How are high/low defined?
For example, I would have found Table 1/2 also better as a graph showing "high" or "low" by arrows, for example.
Figures
17: Figures 1B, 3A are too small and unclear. Figure 5, 6A labels are too small and hard to read.
Image quality needs to be improved + images are distorted
18: Figure 2A: Labeling of x-axis ("NK+mono(patient)" and "NK+mono(healthy)") and labeling of legend with "Healthy"/"Patient" can be misleading. For the x-axis label, it would be better to use "NK+Cancer Monocytes" or "NK+Cancer Patient Monocytes" (same for "Healthy").
19: Figure 3A: Quality! Images very blurred + distorted; also, here misleading labeling as in the previous point (also applies to the following figures).
References
20: The bibliography is inconsistent. The paper contains a high number of self-citations - at least 26 out of 60, almost half, and this is not acceptable.
21: Please use consistent abbreviations throughout the manuscript. If an abbreviation is used for the first time (even in the abstract), please write it out. Abbreviation list should be better arranged alphabetically. Some abbreviations are used that were not previously explained, e.g., LU in M6M, Chapter 2.4. and DI in Chapter 2.5.
22: Revision of the formatting of chapter 3.2, here the text is interrupted.
23: Language and grammar are poor. The entire text must be revised by a native English speaker.